# Phosphate (Pi) Transporter PIT1 Induces Pi Starvation in *Salmonella*-Containing Vacuole in HeLa Cells

**DOI:** 10.3390/ijms242417216

**Published:** 2023-12-07

**Authors:** Wen Yang, Yingxing Feng, Jun Yan, Chenbo Kang, Ting Yao, Hongmin Sun, Zhihui Cheng

**Affiliations:** 1The Key Laboratory of Molecular Microbiology and Technology, Ministry of Education, Nankai University, Tianjin 300071, China; yangwen@mail.nankai.edu.cn (W.Y.); feng_yingxing@163.com (Y.F.); yanjun@mail.nankai.edu.cn (J.Y.); kangchenbo@mail.nankai.edu.cn (C.K.); tingyao@mail.nankai.edu.cn (T.Y.); sunhongmin@mail.nankai.edu.cn (H.S.); 2TEDA Institute of Biological Sciences and Biotechnology, Tianjin Key Laboratory of Microbial Functional Genomics, Nankai University, Tianjin 300457, China; 3Department of Microbiology, College of Life Sciences, Nankai University, Tianjin 300071, China

**Keywords:** *Salmonella enterica* serovar Typhimurium, *Salmonella*-containing vacuoles, Pi starvation, PIT1, *Salmonella* pathogenicity island-2

## Abstract

*Salmonella enterica* serovar Typhimurium (*S*. Typhimurium), an important foodborne pathogen, causes diarrheal illness and gastrointestinal diseases. *S*. Typhimurium survives and replicates in phagocytic and non-phagocytic cells for acute or chronic infections. In these cells, *S*. Typhimurium resides within *Salmonella*-containing vacuoles (SCVs), in which the phosphate (Pi) concentration is low. *S*. Typhimurium senses low Pi and expresses virulence factors to modify host cells. However, the mechanism by which host cells reduce the Pi concentration in SCVs is not clear. In this study, we show that through the TLR4-MyD88-NF-κB signaling pathway, *S*. Typhimurium upregulates PIT1, which in turn transports Pi from SCVs into the cytosol and results in Pi starvation in SCVs. Immunofluorescence and western blotting analysis reveal that after the internalization of *S*. Typhimurium, PIT1 is located on SCV membranes. Silencing or overexpressing PIT1 inhibits or promotes Pi starvation, *Salmonella* pathogenicity island-2 (SPI-2) gene expression, and replication in SCVs. The *S*. Typhimurium Δ*msbB* mutant or silenced TLR4-MyD88-NF-κB pathway suppresses the expression of the SPI-2 genes and promotes the fusion of SCVs with lysosomes. Our results illustrate that *S*. Typhimurium exploits the host innate immune responses as signals to promote intracellular replication, and they provide new insights for the development of broad-spectrum therapeutics to combat bacterial infections.

## 1. Introduction

*Salmonella* is an important foodborne pathogen in all regions of the world [1,2]. *Salmonella* infection is the main cause of the diarrheal illness, which is identified by loose stools and may be accompanied by fever and abdominal cramps [3]. It causes over 180 million (about 9% of all) cases of diarrheal illness and results in 300 thousand (about 41% of all) deaths globally each year [2,4]. *Salmonella* infection also leads to gastrointestinal diseases, especially in young children, the elderly, and immunocompromised patients [5]. *Salmonella enterica* serovar Typhimurium (*S.* Typhimurium) is the most frequent serotype among such diseases [6]. Thus far, antibiotics are the most common tool in the defense against *S.* Typhimurium infection, but it shows increasing drug resistance and new therapeutic methods are in high demand [7].

*S.* Typhimurium’s virulence is dependent on its viability and reproduction inside the host’s cells after infection. *S.* Typhimurium is able to infect macrophages and dendritic, epithelial, and fibroblast-like cells, and it survives and replicates within the *Salmonella*-containing vacuoles (SCVs) to cause acute or chronic infection [8,9]. Although SCVs protect *S.* Typhimurium from the cytosolic bactericidal stress response pathways, *S.* Typhimurium challenges with various stresses in SCVs, including nutrient starvation, reactive oxygen species, and a mildly acidic pH [10,11]. To resist these antibacterial factors, *S.* Typhimurium has evolved efficient survival strategies. *S.* Typhimurium actively regulates SCVs’ biogenesis and the sequential delivery of endolysosomal proteins, through *Salmonella* pathogenicity island-2 (SPI-2)-encoded effectors [12]. SPI-2 effectors generally regulate the movement and maintenance of the SCVs to the juxtanuclear region, and the formation and extension of the *Salmonella*-induced filaments (SIFs) along microtubules [13]. SCV-SIFs enable *S.* Typhimurium to bypass the nutritional limitation in SCVs through obtaining nutrients from the endosomal system [14]. Additionally, the fusion of SCVs with lysosomes is suggested to be completely blocked or delayed, for which neither cathepsins nor mannose 6-phosphate receptor (M6PR) are contracted in SCVs [12,15,16].

To date, the majority of knowledge about the bacterial responses of Pi starvation is obtained from *Escherichia coli* (*E. coli*) [17,18], while the responses of *S.* Typhimurium to Pi limitation are less certain. Recently, a proteomic study of *S.* Typhimurium cultured in MOPS medium suggested that Pi limitation positively regulates SPI-2 genes’ expression, including *ssrA*, *ssrB*, *ssaJ*, *sseA*, *ssaN*, and *sseL* [19]. The SsrAB regulate the majority of the SPI-2 genes’ expression, such as *sifA*, *sseJ*, *sopD2*, and *sifB* [20]. SifA is associated with SCVs’ maturation, SIF formation, and *S.* Typhimurium’s survival and replication [21]. SIFs play crucial roles in providing nutrient access to the bacteria within vacuoles, and mutants lacking SifA are not able to form SIFs and eventually lead to the escape of *Salmonella* from the SCVs [22,23]. In addition, the SifA-SKIP (SifA and kinesin-interacting protein) complex suppresses the recruitment of M6PR and lysosomal enzymes to SCVs [24]. Although it has reported that *S.* Typhimurium encounters Pi limitation in SCVs [19,25], the mechanism by which host cells reduce the Pi concentration in SCVs is not clear.

In this study, we aim to investigate the mechanism behind Pi starvation in SCVs after *S*. Typhimurium infection in HeLa cells. The results show that *S*. Typhimurium exploits the host innate immune responses as signals to regulate virulence genes’ expression, and they provide new insights for the development of broad-spectrum therapeutics to combat pathogenic bacterial infections.

## 2. Results

### 2.1. S. Typhimurium Infection Induces PIT1 Expression in HeLa Cells

In mammalian cells, the transportation of Pi across the cell membrane is mainly dependent on two solute carrier families, SLC20 (including PIT1 and PIT2) and SLC34 (including SLC34A1, SLC34A2, and SLC34A3) [26]. Firstly, we analyzed the transcription levels of these transporters in HeLa cells through quantitative PCR and observed that SLC20 was much more prevalent than SLC34, and PIT1 was the richest among all transporters (Appendix A), indicating that PIT1 may be the main Pi transporter in HeLa cells. Next, to rule out the participation of these genes in *S*. Typhimurium infection, we infected HeLa cells with *S*. Typhimurium for 2 h and analyzed the expression of these genes by qRT-PCR. Only PIT1 displayed 2.13-fold upregulation after *S*. Typhimurium infection (Appendix A). Consistently, western blotting analysis suggested that *S*. Typhimurium infection also induced a 1.88-fold increase in PIT1 compared to the control (Figure 1a). Both the mRNA level and protein level of PIT1 were significantly upregulated in infected cells, indicating that PIT1, but not other transporters, is involved in *S*. Typhimurium infection.

To investigate the distribution of PIT1 in *S*. Typhimurium-infected HeLa cells, we cultured HeLa cells with *S*. Typhimurium–GFP for 2 h and analyzed the colocalization of SCVs with PIT1 through immunofluorescence. The results suggested that 75% of SCVs colocalized with PIT1 (Figure 1b). We also isolated SCVs from *S*. Typhimurium-infected cells at 2 h post infection (p.i.) and observed that PIT1 presented on SCVs (Figure 1c). Collectively, these data indicate that *S*. Typhimurium infection induces PIT1 expression, and PIT1 is located on SCVs after *S*. Typhimurium internalization.

### 2.2. PIT1 Affects S. Typhimurium Intracellular Replication

We then investigated whether PIT1 is associated with *S*. Typhimurium intracellular replication. We transfected with PIT1 siRNA or pcDNA3.1-PIT1 to inhibit or promote PIT1 expression in HeLa cells. Bacterial infection and replication assays suggested that silencing PIT1 (75% reduction, Figure 2a) had no effects on the internalization of *S*. Typhimurium (Figure 2b) but significantly inhibited intracellular replication (Figure 2c). In contrast, overexpressing PIT1 (2.3-fold increase, Figure 2d) had no effects on bacterial invasion (Figure 2e) but significantly promoted bacterial replication (Figure 2f). These data indicate that PIT1 plays an important role in *S*. Typhimurium’s intracellular replication.

### 2.3. PIT1 Transports Pi into Cytosol and Regulates SPI-2 Genes via PhoBR

We next investigated the reason for the importance of PIT1 in *S*. Typhimurium’s intracellular replication. Considering that PIT1 reduces the Pi concentration in bladder epithelial cells [27], we suspected that PIT1 was involved in Pi starvation in SCVs. We purified SCVs at 2 h p.i. from PIT1 siRNA- or pcDNA3.1-PIT1-transfected cells and employed western blotting to analyze PIT1 expression. The results showed that the amount of PIT1 significantly decreased or increased in total cell lysates (TCL) and SCVs from PIT1-silenced (Figure 3a) or -overexpressed (Figure 3b) cells, respectively. We then analyzed the Pi concentrations in SCVs using fluorometric phosphate assays and found that the Pi concentration in PIT1-silenced cells was higher than that in control cells (2.29-fold increase, Figure 3c) but lower in PIT1-overexpressed cells (58% reduction, Figure 3d). These data indicate that upregulated PIT1 transports Pi from SCVs into the cytosol and this leads to Pi starvation in SCVs after *S*. Typhimurium infection.

Considering that low Pi is an important signal for the induction of SPI-2 genes, which are essential for the virulence of *S*. Typhimurium [28,29], we next investigated whether the low Pi induced by PIT1 is enough to regulate SPI-2 genes’ expression. We infected PIT1-silenced or -overexpressed cells with *S*. Typhimurium WT for 2 h and employed qRT-PCR to analyze the mRNA expression of *ssrA*, *ssrB*, and *sifA*. The results suggested that *ssrA*, *ssrB*, and *sifA* were significantly decreased in PIT1-silenced cells (Figure 3e) but increased in PIT1-overexpressed cells (Figure 3f). It is known that *S*. Typhimurium senses Pi limitation through PhoBR [30]; we thus generated a Δ*phoBR* mutant strain of *S*. Typhimurium. We infected PIT1-silenced or -overexpressed cells with *S*. Typhimurium WT or Δ*phoBR* mutant for 2 h and employed qRT-PCR to analyze the expression of these SPI-2 genes. We discovered that the changes in *ssrA*, *ssrB*, and *sifA* resulting from the silencing or overexpression of PIT1 were abolished in the Δ*phoBR* mutant (Figure 3g,h), indicating that the low Pi induced by PIT1 was sufficient to regulate SPI-2 genes’ expression through PhoBR. All these data confirm that the upregulated PIT1 upon *S*. Typhimurium infection reduced the Pi concentration in SCVs, and *S*. Typhimurium senses Pi starvation to regulate SPI-2 genes’ expression.

### 2.4. Upregulated PIT1 Prevents the Fusion of SCVs with Lysosomes

Upon *Salmonella* entry into the host cells, the host immune system recognizes and targets them for lysosomal degradation; however, *Salmonella* can actively regulate the SCVs’ maturation through SPI-2 to escape the cellular lysosomal degradation pathways [31]. Therefore, we next investigated the influence of PIT1 in the fusion of SCVs and lysosomes. Immunofluorescence assays showed that only a small portion of SCVs (<25%) were positive with Cathepsin D (Figure 4a) and M6PR (Figure 4b) at 2 h p.i. in HeLa cells. Silencing PIT1 significantly increased the colocalization of SCVs with Cathepsin D (Figure 4c) and M6PR (Figure 4d), while overexpressing PIT1 slightly reduced these colocalizations (Figure 4e,f). These data indicate that the Pi starvation induced by PIT1 inhibits the recruitment of Cathepsin D and M6PR to SCVs through SPI-2 factors, thus preventing the fusion of SCVs with lysosomes and promoting *S*. Typhimurium’s intracellular replication.

### 2.5. LPS of S. Typhimurium Triggers PIT1 in HeLa Cells

Considering that lipopolysaccharide (LPS), a major membrane component of Gram-negative bacteria, induces PIT1 upregulation in macrophages and BECs [27,32,33], an LPS stimulation assay was employed and the results showed that LPS from *S*. Typhimurium strongly induced PIT1 expression (Figure 5a). For further investigation, we constructed a Δ*msbB* mutant strain that lacked one acyl chain and found that the PIT1 upregulation was impaired in Δ*msbB*-infected cells (Figure 5b). In addition, the Δ*msbB* mutant showed the typical internalization (Figure 5c), deficient intracellular replication (Figure 5d), and decreased PIT1 in TCL and SCVs (Figure 5e), and the greater colocalization of SCVs with Cathepsin D (Figure 5f) and M6PR (Figure 5g). Collectively, these results indicate that LPS in *S*. Typhimurium induces PIT1 expression after infection, which promotes the Pi starvation in SCVs and avoids the fusion of SCVs with lysosomes.

### 2.6. S. Typhimurium Exploits TLR4-MyD88-NF-kB Pathway to Regulate PIT1

Considering that Toll-like receptor 4 (TLR4) senses LPS, and the TLR4-MyD88-NF-kB pathway plays a significant role in innate immunity [34], we then investigated whether the upregulation of PIT1 is mediated by the TLR4-MyD88-NF-kB pathway. The silencing of TLR4, MyD88, or p65 (62% reduction for TLR4, 65% reduction for MyD88, and 91% reduction for p65, Appendix A) resulted in significant reductions in PIT1 in TCL and SCVs (Figure 6a–c) and SPI-2 genes’ expression (Figure 6d–f), but an obvious promotion of the colocalization of SCVs with Cathepsin D (Figure 6g) and M6PR (Figure 6h). The above data indicate that *S*. Typhimurium exploits the host innate immune pathway, which leads to Pi starvation in SCVs, to promote the SPI-2 genes’ expression and inhibit the fusion of SCVs and lysosomes.

## 3. Discussion

In this study, we evaluated the mechanism behind host cell-induced Pi starvation in SCVs in HeLa cells. Pi limitation is a widespread host defense in response to bacterial infection, since many pathogens encounter Pi starvation after infection [27,35]. In *E. coli*, *Salmonella*, and *Shigella flexneri* (*S*. *flexneri*), Pi limitation exists in the bacteria-containing vesicles within host cells and is of great importance in bacterial intracellular survival and replication [19,27,35]. Recently, it has been reported that the host PIT1 induces Pi starvation in uropathogenic *E. coli* (UPEC)-containing vesicles [27]; however, little is known about *Salmonella* and other pathogens. Here, we found that PIT1 is the main reason for the reduction in Pi in SCVs. After the internalization of *S*. Typhimurium in HeLa cells through endocytosis, the membrane of the SCV departs from the host cell membrane [36]; thus, the phosphate transporters travel on the membranes of these vacuoles. We illustrated that, in HeLa cells, PIT1 presented on SCVs and colocalized with *S.* Typhimurium. Silencing or overexpressing PIT1 inhibited or promoted the Pi reduction in SCVs, which suggests that PIT1 transports Pi from SCVs into the cytosol and results Pi starvation in SCVs. Further, we also demonstrated that host cells activated the innate immune responses through the TLR4-MyD88-NF-κB pathway to promote PIT1 expression upon *S.* Typhimurium infection. For bacterial pathogens, LPS usually functions as an immunomodulatory molecule to elicit efficient innate immune responses through TLR4-MyD88-NF-κB and induce the expression of inflammatory cytokines in the host [37]. The Δ*msbB* mutant or the silencing of the TLR4-MyD88-NF-κB pathway inhibited PIT1 expression in TCL and SCVs and impaired *S*. Typhimurium’s intracellular replication. Therefore, these results confirm that PIT1 is the major Pi transporter in both BECs and HeLa cells, suggesting that PIT1 is also able to function in other epithelial cells, and this might provide specific targets for new therapeutics for bacterial infections.

Pi is of great importance for living organisms and is indispensable in nucleic acids, membrane lipids, proteins, and carbohydrates [38]. Sensing environmental Pi is a crucial step for bacterial infection. In *E*. *coli* and most other bacterial species, Pi starvation is sensed by PhoBR [39]. Here, we found that silencing or overexpressing PIT1 inhibited or promoted SPI-2 genes’ expression, and the Δ*phoBR* mutant significantly impaired the influence of Pi limitation on SPI-2 genes’ expression compared with the *S*. Typhimurium WT, which suggested that PhoBR senses the Pi limitation and regulates SPI-2 genes’ expression. However, PhoBR is not the only Pi signal transduction mechanism in bacteria. The TCSs Spo0B–Spo0A, the transcription regulators AbrB and ScoC, and the teichoic acid synthesis pathway are also suggested to be involved in Pi signal detection and signaling [40,41,42]. In addition, Pi limitation induces various genes in different bacteria, including the high-affinity Pst system, the *phn* operon, and the *upg* operon in *E. coli* [43], related to the cell composition, and it reallocates the assimilated Pi in *Bacillus subtilis* [44] and virulence genes in *S*. *flexneri* [35]. Whether there are other proteins that are important for *S*. Typhimurium’s detection of the environmental Pi is still unclear, and the other genes induced by Pi limitation also need to be studied in the future.

Pathogens have evolved sophisticated mechanisms enabling them to survive and proliferate in eukaryotic cells. The intracellular bacterial pathogens are mainly divided into ‘vacuolar’ and ‘cytosolic’ [45]. *S*. Typhimurium, belonging to the first group, prefers to modify the structure of the vacuolar membrane and maintains its integrity [46]. In SCVs, *S*. Typhimurium is able to manipulate the maturation of SCVs, limit fusion with lysosomes, and ultimately establish a replicative niche in host cells [47]. In the present study, we found that after *S*. Typhimurium infection, only approximately a quarter of SCVs were positive with Cathepsin D or M6PR. Silencing or overexpressing PIT1 promoted or suppressed the recruitment of Cathepsin D and M6PR. Together with this, Pi starvation promoted SPI-2 genes’ expression, and SifA, as well as other SPI-2 effectors, is involved in suppressing the recruitment of M6PR and lysosomal enzymes to SCVs [24,48,49]. We found that PIT1 induced Pi starvation in SCVs to not only promote *S*. Typhimurium’s virulence but also contribute to avoiding the fusion of SCVs and lysosomes. Additionally, this was further proven by the Δ*msbB* mutant or silencing of the TLR4-MyD88-NF-κB pathway, which inhibited PIT1 expression on SCVs but promoted the recruitment of Cathepsin D and M6PR to SCVs. These data confirm that *S*. Typhimurium exploits the innate immune responses to modulate the maturation of SCVs for their intracellular survival and replication. On the other hand, Pi starvation activates *pldA* expression to degrade phospholipids on the membrane and disrupt the fusiform vesicles, and it promotes UPEC’s escape into the cytosol to facilitate cytosolic survival [27]. Although the vacuolar Pi concentrations change similarly after bacterial infection, Pi starvation induces different survival strategies. It is possible that different bacteria have acquired different styles, and this needs further study in the future.

In conclusion, these data enrich our knowledge of the mechanism behind the host reduction of Pi in SCVs and how *S*. Typhimurium exploits Pi starvation as a signal to survive and replicate in SCVs, which may provide specific targets for new therapeutics for bacterial infections.

## 4. Materials and Methods

### 4.1. Bacterial Strains

*Salmonella enterica* serovar Typhimurium strain ATCC 14028 was designated as *S*. Typhimurium. The Δ*phoBR* and Δ*msbB* mutant strains were constructed using the *λ*-Red recombinase system by replacing the target genes with a chloramphenicol or kanamycin resistance gene cassette, respectively, as described previously [50]. The *S*. Typhimurium WT and Δ*msbB*, which express green fluorescent protein (GFP), were constructed through electrotransformation, as described previously [51]. *S*. Typhimurium was cultured in Luria–Bertani (LB) medium at 37 °C to the stationary phase (OD_600_ = 1.0, ~10^9^ CFUs/mL). The medium was supplemented with chloramphenicol (25 μg/mL), gentamycin (20 or 100 μg/mL), kanamycin (50 μg/mL), or ampicillin (100 μg/mL) (all from Sangon Biotech, Shanghai, China, A600118, A620217, A600286, A610028) when needed.

The bacterial strains and plasmid used in this study are listed in Appendix A. The primers used in this study are listed in Appendix A.

### 4.2. Antibodies

The antibodies used in this study and the relevant information are as follows: rabbit polyclonal to PIT1 (Abcam, Cambridge, UK, ab237527), GAPDH (Sangon Biotech, D110016), and LAMP2A (Abcam, ab18528); rabbit monoclonal to Rab7 (Abcam, ab137029), Cathepsin D (Abcam, ab75852), M6PR (Abcam, ab124767), MyD88 (Abcam, ab133739), and p65 (Abcam, ab32536); mouse monoclonal to GroEL (Abcam, ab82592), and TLR4 (Santa Cruz Biotechnology, Santa Cruz, CA, USA, sc-13593); Goat Anti-Rabbit IgG H&L-conjugated Alexa Fluor 647 IgG (Abcam, ab150083), Goat Anti-Mouse IgG (BBI, Shanghai, China, D111024), and Goat Anti-Rabbit IgG (BBI, D110058, D110058). The antibodies were diluted at 1:1000 for western blotting analysis and at 1:300 for immunofluorescence analysis.

### 4.3. Cell Culture and Transfection

HeLa cells were cultured in Dulbecco’s Modified Eagle’s Medium (DMEM) supplemented with 10% fetal bovine serum (defined as DMEM cell culture) at 37 °C with 5% CO_2_.

For the silencing of proteins, cells were transfected with control siRNA or specific siRNA using Lipofectamine RNAi MAX (Invitrogen, Waltham, MA, USA, 13778150) according to the manufacturer’s instructions. Specific siRNAs (Control siRNA, PIT1 siRNA, TLR4 siRNA, MyD88 siRNA, and p65 siRNA) were all from RiBoBio.

For the overexpression of the PIT1 protein, the coding region of the PIT1 gene was cloned into the pcDNA3.1 plasmid between the BamHⅠ and XhoI sites, and the constructed plasmids were extracted using the EndoFree Mini Plasmid Kit (Tiangen, Beijing, China, DP118). Cells were transiently transfected with pcDNA3.1-PIT1 (pcDNA3.1 was used as the control) using Lipofectamine 3000 (Invitrogen, L3000015, USA) according to the manufacturer’s instructions.

After 48 h transfection, the cells were collected directly or infected with bacteria.

### 4.4. RNA Extraction and qRT-PCR

To investigate the expression patterns of Na/Pi-cotransporter genes, HeLa cells were collected directly or infected with or without *S*. Typhimurium for 2 h. To determine the expression of *S*. Typhimurium SPI-2 genes, cells were collected at 2 h p.i. Total RNA was extracted using TRIzol Reagent (Invitrogen, 15596026) according to the manufacturer’s instructions. Purified RNA was reverse-transcribed using the PrimeScript™ Realtime PCR Kit (Takara, Shiga, Japan, RR036A). qRT-PCR was carried out using SYBR Green PCR Master Mix (Applied Biosystems, Waltham, MA, USA, 4367659,) on an ABI PRISM 7500 Sequence Detection System (Thermo Fisher, Waltham, MA, USA). For relative mRNA expression, the human GAPDH or *S*. Typhimurium 16S rRNA was used as the reference, respectively. The comparative Ct method (2^−ΔΔCt^ method) was utilized to analyze the relative mRNA expression [52].

### 4.5. Quantitative PCR Analysis

Quantitative PCR analysis was performed as previously described, with minor modifications [53]. The corresponding target DNA fragment of each gene was amplified from the cDNA of HeLa cells. The DNA fragment was cloned into the pMD19-T Vector (Takara, Shiga, Japan, 6013) to serve as a standard. The standard curve was generated from a 10-fold serial dilution (from 10^2^ to 10^9^ copies) of the plasmid, and only the linear regression R-values above 99% were accepted. The plasmid copy number was calculated based on the molecular size of the plasmid and the concentration of the plasmid. The cDNA copy number in samples was calculated based on the standard curve. Quantitative PCR was performed as qRT-PCR. For each quantitative PCR analysis, the dissociation curve was determined to ensure the absence of primer dimers.

### 4.6. Bacterial Infection and Replication Assays

Bacterial infection and replication assays were performed as previously described, with minor modifications [54]. *S*. Typhimurium was grown overnight and then subcultured in LB to the stationary phase. *S*. Typhimurium was harvested using a centrifuge, resuspended, and opsonized in DMEM cell culture at 37 °C for 30 min. HeLa cells were washed with PBS three times and incubated with *S*. Typhimurium at a multiplicity of infection (MOI) of 10:1. After incubation for 50 min (T = 0), HeLa cells were washed with PBS three times and cultured in a DMEM cell culture supplemented with gentamycin (100 μg/mL) for 1 h at 37 °C to kill the extracellular bacteria. Then, the cells were cultured in a DMEM cell culture supplemented with 20 μg/mL gentamycin was used for the rest of the experiments. For intracellular replication assay, the cells were lysed using 0.1% Triton X-100 (Sigma, Shanghai, China, 9036-19-5) at 2 h p.i. and 16 h p.i., and intracellular bacteria were applied in a 10-fold serial dilution and placed onto LB agar. The fold of intracellular replication was calculated as the ratio of the CFUs at 16 h p.i. relative to that at 2 h p.i.

### 4.7. Immunofluorescence Analysis

The immunofluorescence analysis was performed as previously described, with minor modifications [55]. After infection with *S*. Typhimurium–GFP at an MOI of 10:1 for 2 h, cells were washed with PBS three times and fixed in 4% cold PFA (Beyotime, Shanghai, China, P0099) for 15 min. Then, the cells were incubated with 0.1% Triton X-100 for permeabilization and blocked with 3% BSA (BBI, D110078) at 25 °C for 1 h. The cells were incubated with the indicated primary antibodies (1:300 dilution in 3% BSA) at 4 °C overnight. The cells were then washed with PBST (0.1% Tween-20 in PBS) three times and incubated with Goat Anti-Rabbit IgG H&L-conjugated Alexa Fluor^®^ 647 IgG (1:1000 in 3% BSA) at 25 °C for 2 h. The cells were mounted on ProLong™ Gold and Diamond Antifade Mountants (Invitrogen, MP36930). Images were visualized using the Carl Zeiss LSM 800 (Carl Zeiss AG, Oberkochen, Germany). Three slides were examined for each group, and about 100 intracellular *S*. Typhimurium were counted on each slide.

### 4.8. Isolation of Intracellular SCVs

The intracellular SCVs were purified as described previously, with minor modifications [27]. Briefly, BioMag Carboxyl Magnetite Particles (Bangs Laboratories, Fishers, IN, USA, BM570) were washed with 0.1 M MES buffer (2-(N-morpholino) ethanesulphonic acid, pH 5.2) three times and incubated with EDAC (1-ethyl-3-(3-dimethylaminopropyl) carbodiimide) for 30 min for activation. The activated particles were incubated with 1 × 10^8^ live *S*. Typhimurium colonies at 25 °C for 1 h. The mixtures were washed and resuspended in 1% BSA to block the reactive sites of the particles at 25 °C for 30 min. HeLa cells were infected with the magnetic bacteria. At 2 h p.i., the infected cells were washed with cold PBS three times and passed through a 30-gauge dental needle. The cell lysates were centrifuged at 200× *g* for 5 min at 4 °C to remove the intact cells. The supernatant was transferred to a magnetic rack on ice for 15 min. The isolated SCVs were washed with cold PBS, boiled in 1× SDS loading buffer, and analyzed using western blotting analysis.

### 4.9. Western Blotting Analysis

Proteins were separated in 12% SDS-PAGE and transferred onto a polyvinylidene difluoride membrane (Millipore, Darmstadt, Germany, IEVH85R). The membranes were blocked in Western Blocking Buffer (Beyotime, P0252, China) and incubated with specific primary antibodies (1:1000 dilution in 5% non-fat milk) at 4 °C overnight, and then washed with TBST (0.1% Tween-20 in TBS) three times. The membrane was then incubated with HRP-conjugated Goat Anti-Mouse or Rabbit IgG (1:5000, BBI, D111024 or D110058, respectively). Finally, the bands were visualized using the SuperSignal West Pico Chemiluminescent Substrate (Thermo, 34580, USA). The protein expression was analyzed using ImageJ (NIH, https://imagej.nih.gov, accessed on 14 December 2022). Human GAPDH or bacterial GroEL was used to show that similar numbers of infected cells or SCVs were employed, respectively.

### 4.10. Determination of the Phosphate Concentration in SCVs

The Pi concentration in the isolated SCVs was analyzed using a Fluorimetric Phosphate Assay Kit (AAT Bioquest, Pleasanton, CA, USA, 21660). Briefly, the extracted SCVs were lysed with water and centrifuged at 12,000 rpm, 4 °C, for 10 min. The supernatants were collected, diluted, and incubated with Analysis Buffer and Phosphate Working Solution for 30 min at 25 °C. The absorbance was measured at 540 nm using a Spark 10M (Tecan, Männedorf, Switzerland, 1611011065). Blank wells were determined as zero. The standard curve was prepared following the instructions at the same time.

### 4.11. LPS Stimulation

HeLa cells were washed with PBS three times and then cultured in a DMEM cell culture supplemented with or without LPS (100 ng/mL, Sigma, L7895) at 37 °C with 5% CO_2_. After incubation for 2 h, HeLa cells were washed with PBS three times and lysed in RIPA buffer (supplemented with PMSF) on ice for 30 min, centrifuged at 12,000 rpm, 4 °C, for 10 min, and then boiled with SDS loading buffer at 99 °C for 10 min. The samples were used for western blotting.

### 4.12. Quantification and Statistical Analysis

Statistical significance was analyzed with the GraphPad Prism 9 software (GraphPad Inc., San Diego, CA, USA) using the Student’s *t* test, one-way ANOVA, or two-way ANOVA (as stated in the figure legends). All experiments were repeated at least three times independently. Data indicate means ± SD (*n* = 3). *p*-values less than 0.05 were considered statistically significant.

## Figures and Tables

**Figure 1 ijms-24-17216-f001:**
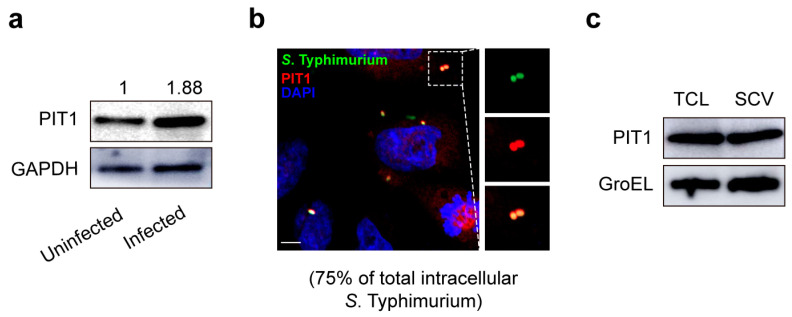
*S*. Typhimurium infection induces PIT1 expression in HeLa cells. (**a**) Western blotting analysis to show the expression of PIT1 at 2 h after *S*. Typhimurium infection. The expression level is indicated at the top of the blots. (**b**) Immunofluorescence assay to show colocalization of PIT1 with *S*. Typhimurium. Scale bar = 5 μm. The number indicates the percentage of SCVs colocalized with PIT1 relative to total intracellular SCVs. (**c**) Western blotting analysis to show the presence of PIT1 on SCVs.

**Figure 2 ijms-24-17216-f002:**
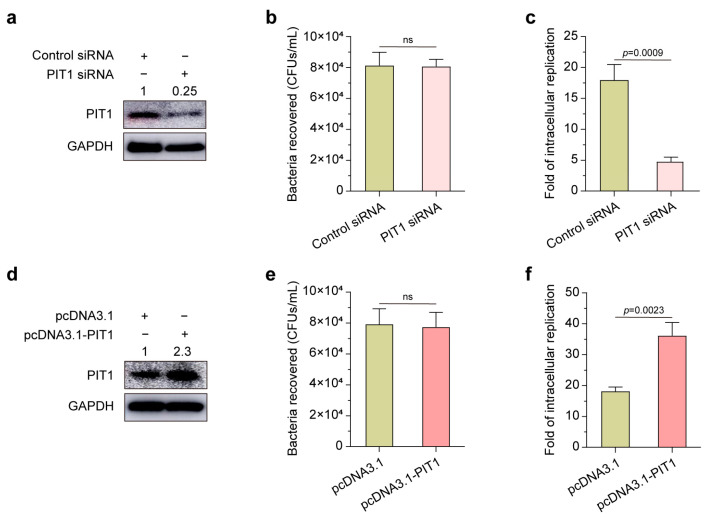
PIT1 affects *S*. Typhimurium’s intracellular replication. (**a**) Western blotting analysis to show the silencing efficiency of PIT1 siRNA in HeLa cells. HeLa cells were transfected with control siRNA or PIT1 siRNA. The silencing efficiency is indicated at the top of the blots. (**b**,**c**) The bar graphs show CFUs of bacteria recovered at 2 h p.i. (**b**), and the fold of intracellular replication at 16 h p.i. (**c**) in control siRNA- or PIT1 siRNA-transfected cells. (**d**) Western blotting analysis to show the overexpression efficiency of pcDNA3.1-PIT1 in HeLa cells. HeLa cells were transfected with pcDNA3.1 or pcDNA3.1-PIT1. The overexpression efficiency is indicated at the top of the blots. (**e**,**f**) The bar graphs show CFUs of bacteria recovered at 2 h p.i. (**e**) and the fold of intracellular replication at 16 h p.i. (**f**) in pcDNA3.1- or pcDNA3.1-PIT1-transfected cells. Data indicate means ± SD (*n* = 3). The significant differences are represented by *p*-values determined with Student’s *t* test. ns, nonsignificant.

**Figure 3 ijms-24-17216-f003:**
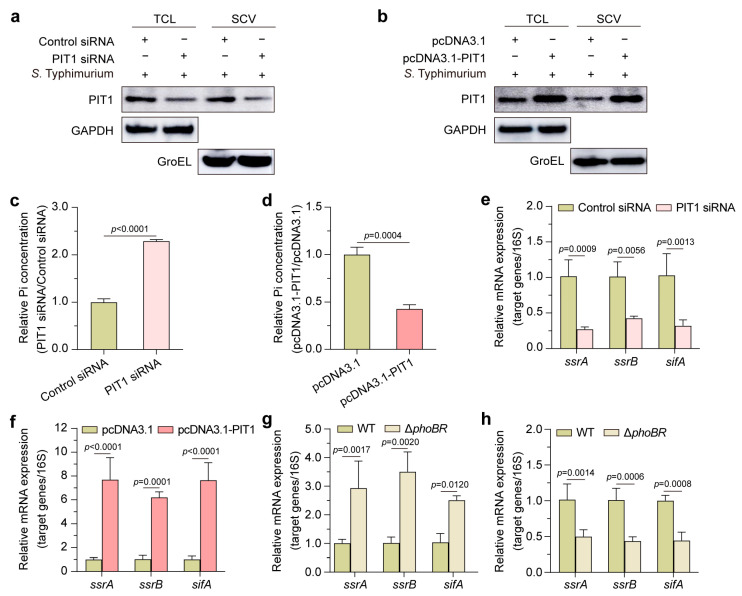
PIT1 transports Pi into cytosol and regulates SPI-2 genes via PhoBR. (**a**,**b**) Western blotting analysis to show the expression of PIT1 in TCL and SCVs from PIT1 siRNA (**a**) or pcDNA3.1-PIT1 (**b**) transfected cells. (**c**,**d**) The relative Pi concentrations in SCVs were determined with fluorimetric phosphate assay kit. The bar graphs show the Pi concentration in SCVs transfected with PIT1 siRNA (**c**) or pcDNA3.1-PIT1 (**d**) relative to those of control. (**e**) qRT-PCR analysis of the relative mRNA expression of *ssrA*, *ssrB*, and *sifA* from *S*. Typhimurium at 2 h p.i. in control siRNA- or PIT1 siRNA-transfected cells. (**f**) qRT-PCR analysis of the relative mRNA expression of *ssrA*, *ssrB*, and *sifA* from *S*. Typhimurium at 2 h p.i. in pcDNA3.1- or pcDNA3.1-PIT1-transfected cells. (**g**,**h**) qRT-PCR analysis of the relative mRNA expression of *ssrA*, *ssrB*, and *sifA* from wild-type strain (WT) or Δ*phoBR* mutant at 2 h p.i. in PIT1 siRNA (**g**) or pcDNA3.1-PIT1 (**h**) transfected cells. Data indicate means ± SD (*n* = 3). The significant differences are represented by *p*-values determined with Student’s *t* test (**c**,**d**) or two-way ANOVA (**e**–**h**).

**Figure 4 ijms-24-17216-f004:**
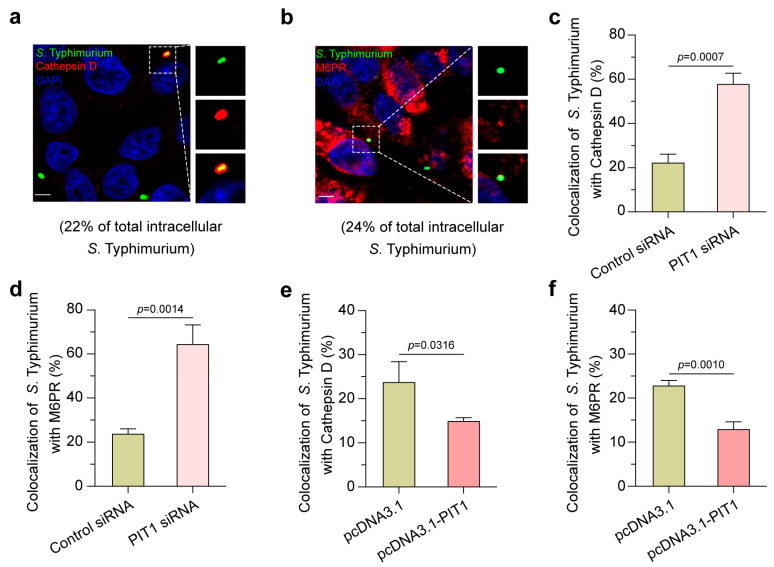
Upregulated PIT1 prevents the fusion of SCVs with lysosomes. (**a**,**b**) Immunofluorescence assays to show colocalization of *S*. Typhimurium with Cathepsin D (**a**) and M6PR (**b**). Scale bar = 5 μm. The number indicates the percentage of SCVs colocalized with PIT1 relative to total intracellular SCVs. (**c**,**d**) The bar graphs show the percentage of *S*. Typhimurium localized with Cathepsin D (**c**) and M6PR (**d**) in control siRNA- or PIT1 siRNA-transfected cells. (**e**,**f**) The bar graphs show the percentage of *S*. Typhimurium localized with Cathepsin D (**e**) or M6PR (**f**) in pcDNA3.1- or pcDNA3.1-PIT1-transfected cells. Data indicate means ± SD (*n* = 3 slides). The significant differences are represented by *p*-values determined with Student’s *t* test.

**Figure 5 ijms-24-17216-f005:**
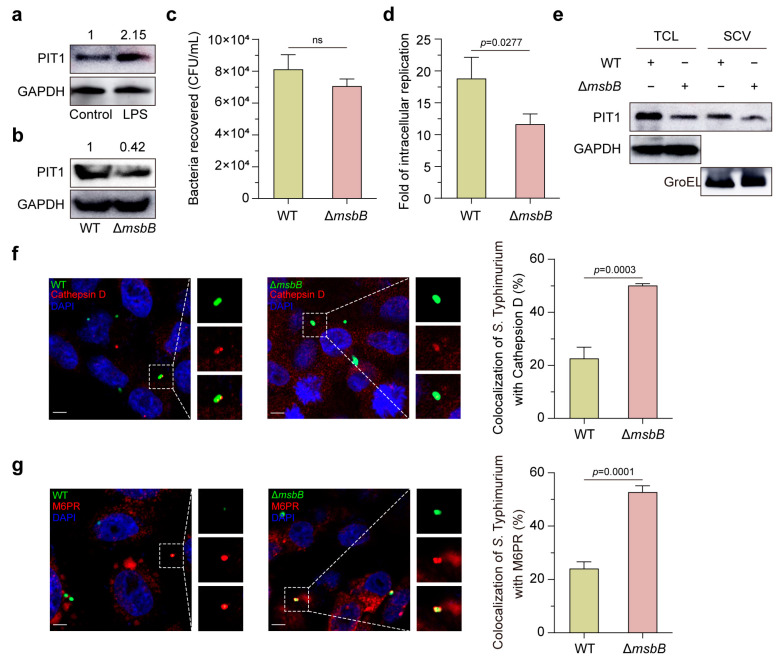
LPS of *S*. Typhimurium triggers PIT1 in HeLa cells. (**a**,**b**) Western blotting analysis to show the expression of PIT1 in HeLa cells after LPS stimulation (**a**) or Δ*msbB* mutant infection (**b**). The expression level is indicated at the top of the blots. (**c**,**d**) The bar graphs show CFUs of WT or Δ*msbB* mutant recovered at 2 h p.i. (**c**), and the fold of intracellular replication at 16 h p.i. (**d**) in HeLa cells. (**e**) Western blotting analysis to show the expression of PIT1 in TCL and SCVs in WT or Δ*msbB*-infected cells. (**f**,**g**) Immunofluorescence assays to show colocalization of WT or Δ*msbB* mutant with Cathepsin D (**f**) and M6PR (**g**). Scale bar = 5 μm. Data indicate means ± SD (*n* = 3 slides). The significant differences are represented by *p*-values determined with the Student’s *t* test. ns, nonsignificant.

**Figure 6 ijms-24-17216-f006:**
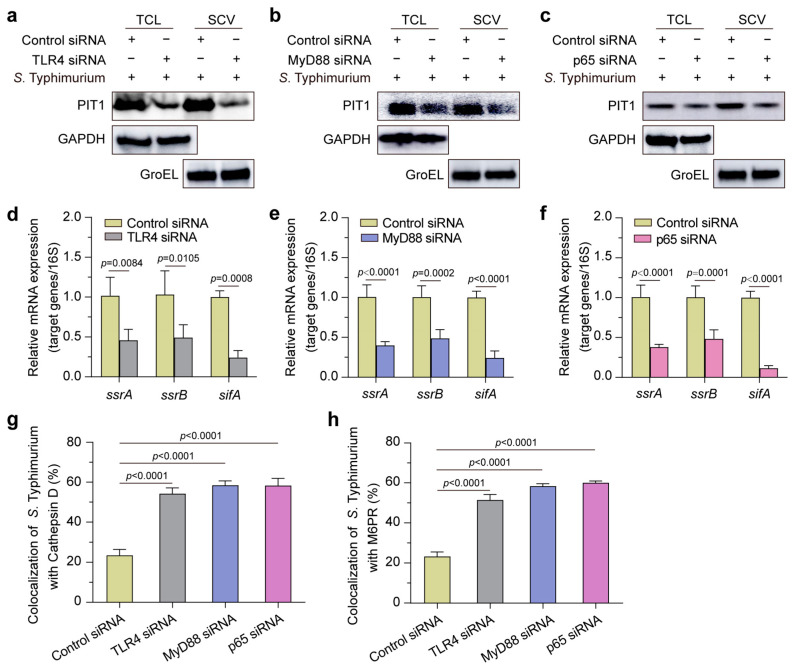
*S*. Typhimurium exploits TLR4-MyD88-NF-κB pathway to regulate PIT1. (**a**–**c**) Western blotting analysis to show expression level of PIT1 in TCL and SCVs in TLR4 (**a**), MyD88 (**b**), or p65 (**c**) siRNA-transfected cells. (**d**–**f**) qRT-PCR analysis of the relative mRNA expression of *ssrA*, *ssrB*, and *sifA* in TLR4 (**c**), MyD88 (**d**), or p65 (**e**) siRNA-transfected cells. (**g**,**h**) Immunofluorescence assays to show colocalization of SCVs with Cathepsin D (**g**) and M6PR (**h**) in TLR4, MyD88, or p65 siRNA-transfected cells. Data indicate means ± SD (*n* = 3). The significant differences are represented by *p*-values determined with two-way ANOVA (**d**–**f**) or one-way ANOVA (**g**,**h**).

## Data Availability

Data are contained within the article and Appendix A.

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
