# Peer review of "Phosphate (Pi) Transporter PIT1 Induces Pi Starvation in Salmonella-Containing Vacuole in HeLa Cells"

_ijms, 2023, doi:10.3390/ijms242417216_

Round 1

Reviewer 1 Report

Comments and Suggestions for Authors

The article titled “Phosphate (Pi) transporter PIT1 induces Pi starvation in Salmonella-containing vacuole in HeLa cells” is a valuable paper where the authors present evidence that S. Typhimurium, via the TLR4–MyD88–NF-κB signaling pathway, upregulates PIT1. PIT1, in turn, facilitates the transport of Pi from SCVs to the cytosol, inducing Pi starvation within SCVs. Silencing or overexpressing PIT1 respectively inhibits or promotes Pi starvation, the expression of SPI-2 genes, and replication within SCVs. Furthermore, the authors indicate that an S. Typhimurium ΔmsbB mutant or the silencing of the TLR4–MyD88–NF-κB pathway leads to the suppression of SPI-2 gene expression and facilitates the fusion of SCVs with lysosomes. The authors also elucidate how S. Typhimurium leverages host innate immune responses as signals to enhance intracellular replication. These insights contribute to the understanding of bacterial infection mechanisms and offer valuable perspectives for the development of broad-spectrum therapeutics to combat such infections.

It is very valuable work. I have only minor comments.

Line 59: put whole name and after abbreviation Escherichia coli (E. coli)

Line 231: E. coli

Line 319; 349;  351; 373: “p.i.” what this abbr. mean ?

All the best and Congrats to the Authors

Author Response

Response to Reviewer 1 Comments

Dear Reviewer,

We thank you very much for your careful review. We have corrected our manuscript carefully, and we believe, by doing so, we have improved and strengthened the manuscript substantially. Please find the detailed responses below and the corresponding corrections highlighted in the re-submitted files

Are all the cited references relevant to the research?

Yes

Is the research design appropriate?

Yes

Are the methods adequately described?

Yes

Are the results clearly presented?

Yes

Are the conclusions supported by the results?

Yes

Point-by-point response to Comments and Suggestions for Authors

The article titled “Phosphate (Pi) transporter PIT1 induces Pi starvation in Salmonella-containing vacuole in HeLa cells” is a valuable paper where the authors present evidence that S. Typhimurium, via the TLR4–MyD88–NF-κB signaling pathway, upregulates PIT1. PIT1, in turn, facilitates the transport of Pi from SCVs to the cytosol, inducing Pi starvation within SCVs. Silencing or overexpressing PIT1 respectively inhibits or promotes Pi starvation, the expression of SPI-2 genes, and replication within SCVs. Furthermore, the authors indicate that an S. Typhimurium ΔmsbB mutant or the silencing of the TLR4–MyD88–NF-κB pathway leads to the suppression of SPI-2 gene expression and facilitates the fusion of SCVs with lysosomes. The authors also elucidate how S. Typhimurium leverages host innate immune responses as signals to enhance intracellular replication. These insights contribute to the understanding of bacterial infection mechanisms and offer valuable perspectives for the development of broad-spectrum therapeutics to combat such infections.

It is very valuable work. I have only minor comments.

Comments 1: Line 59: put whole name and after abbreviation Escherichia coli (E. coli)

Response 1: Thanks for the suggestion. We agree with this comment. Therefore, we have corrected E. coli in line 59 in the original manuscript as Escherichia coli (E. coli), which can be found in the revised manuscript in line 59.

Comments 2: Line 319; 349; 351; 373: “p.i.” what this abbr. mean ?

Response 2: Thanks for the comment. We agree with this, and Escherichia. coli in line 59 in the original manuscript has been corrected as E. coli, which can be found in the revised manuscript in line 231.

Comments 3: Line 59: put whole name and after abbreviation Escherichia coli (E. coli)

Response 3: Thanks for the comment. “p.i.” means post infection, and it has been defined at the first appearance in line 101 in the revised manuscript.

Reviewer 2 Report

Comments and Suggestions for Authors

Dear authors

I hope all of you are always fine. Regarding the revision of the manuscript No. ijms-2716905, titled “Phosphate (Pi) transporter PIT1 induces Pi starvation in Salmonella-containing vacuole in HeLa cells”. Really it is a very interesting study; however, some comments should be replied.

Minor comments

1-    Lines 71-79 should include the aim of the study only. No results or conclusions are mentioned here as they are mentioned only at the end of abstract and discussion. Please correct.

2-    Please clarify the specificity of the used tests to PIT1 only without other transporters involved in S. Typhimurium infection.

3-    Figure S2b should be added to figure 2.

4-    The whole results obtained are very interesting and should be discussed in more details. So, please add one or 2 more paragraph to your discussion.

Author Response

Response to Reviewer 2 Comments

Dear Reviewer,

We thank you very much for your careful review. We have heeded the suggestions and have added additional discussion to clarify and strengthen our findings. We believe, by doing so, we have improved and strengthened the manuscript substantially. Please find the detailed responses below and the corresponding corrections highlighted in the re-submitted files

Are all the cited references relevant to the research?

Yes

Is the research design appropriate?

Yes

Are the methods adequately described?

Yes

Are the results clearly presented?

Yes

Are the conclusions supported by the results?

Yes

Point-by-point response to Comments and Suggestions for Authors

Dear authors

I hope all of you are always fine. Regarding the revision of the manuscript No. ijms-2716905, titled “Phosphate (Pi) transporter PIT1 induces Pi starvation in Salmonella-containing vacuole in HeLa cells”. Really it is a very interesting study; however, some comments should be replied.

Minor comments

Comments 1: Lines 71-79 should include the aim of the study only. No results or conclusions are mentioned here as they are mentioned only at the end of abstract and discussion. Please correct.

Response 1: Thanks for the suggestion. We agree with this comment. Therefore, we have corrected lines 71-79 in the original manuscript as “In this study, we aim to investigate the mechanism under Pi starvation in SCVs after S. Typhimurium infection in HeLa cells. The results showed that S. Typhimurium exploits host innate immune responses as signals to regulate virulence genes expression and provide new insights on development of broad spectrum therapeutic to combat pathogenic bacterial infection.”, which can be found in the revised manuscript in lines 71-75.

Comments 2: Please clarify the specificity of the used tests to PIT1 only without other transporters involved in S. Typhimurium infection.

Response 2: Thanks for the comment. The reasons for the specificity of the used tests to PIT1 only without other transporters involved in S. Typhimurium infection are listed as follows:

Firstly, the results from quantitative PCR in HeLa cells suggested that the enrichment of SLC20 (including PIT1 and PIT2) is much higher than that in SLC34 (including SLC34A1, SLC34A2, and SLC34A3), and this is consistent with previously reported that PIT1 and PIT2 are expressed ubiquitously in all tissues, and the other three transporters are present in the kidneys and the gut (PMID: 18305372).

Secondly, the results from qRT-PCR also suggested that only PIT1 was significantly upregulated after S. Typhimurium infection, which is consistent with previously reported that PIT1 is the major Pi transporter in HeLa cells (PMID: 29233890, and 34160116).

Comments 3: Figure S2b should be added to figure 2

Response 3: Thanks for the comments. Considering your valuable suggestion, we have moved all figures in Figure S2 in the original manuscript into Figure 2 (Figure 2a and Figure 2d in the revised manuscript) for a better clarification. We have moved Figure 2a and 2b in the original manuscript to Figure 2b and 2c in the revised manuscript. We have moved Figure 2c and 2d in the original manuscript to Figure 2e and 2f in the revised manuscript. We have moved Figure S3 in the original manuscript to Figure S2 in the revised manuscript.

Comments 4: The whole results obtained are very interesting and should be discussed in more details. So, please add one or 2 more paragraph to your discussion.

Response 3: Thank you for the comments. We have discussed the results obtained in the present study more in discussion. The detailed information are as follows:

(1) Here, we found PIT1 is the main reason for the reduction of Pi in SCVs. After the internalization of S. Typhimurium in HeLa cells through endocytosis, the membrane of SCVs departs from the host cell membrane (PMID: 11207539), thus, the phosphate transporters traveled on the membrane of these vacuoles. We illustrated that, in HeLa cells, PIT1 presented on SCVs and colocalized with S. Typhimurium. Silencing or overexpressing PIT1 inhibited or promoted Pi reduction in SCVs, which suggested that PIT1 transport Pi from SCVs into the cytosol, and resulted Pi starvation in SCVs. (lines 236-242 in the revised manuscript).

(2) ΔphoBR mutant significantly impaired the influence of Pi limitation on SPI-2 genes expression compared with S. Typhimurium WT, which suggested that PhoBR sense Pi limitation and regulate SPI-2 genes expression (lines 257-259 in the revised manuscript).

(3) In SCVs, S. Typhimurium is able to manipulate the maturation of SCVs, limit fusion with lysosomes and ultimately establish a replicative niche in host cells (PMID: 15036145). In the present study, we found that after S. Typhimurium infection, only approximately a quarter SCVs was positive with Cathepsin D or M6PR. Silencing or overexpressing PIT1promoted or suppressed the recruitment of Cathepsin D and M6PR. Together with that Pi starvation promoted SPI-2 genes expression, and SifA, as well as other SPI-2 effectors are involved in suppressing the recruitment of M6PR and lysosomal enzymes to SCVs (PMID: 23162002, 26299973, and 16177296), we found that PIT1 induced Pi starvation in SCVs not only promote S. Typhimurium virulence, but also contribute in avoiding the fusion of SCVs and lysosomes. Additionally, this is further proved by ΔmsbB mutant or silencing of TLR4-MyD88-NF-κB pathway, which inhibit PIT1 expression on SCVs, but promote the recruitment of Cathepsin D and M6PR to SCVs (lines 272-283 in the revised manuscript).
